# Clinical epidemiology of COVID-19 among hospitalized children in rural western Kenya

**Adino Tesfahun Tsegaye**[1]*, **Christina Sherry**[2], **Chrisantus Oduol**[3], **Joyce Otieno**[3], **Doreen Rwigi**[3], **Mary Masheti**[3], **Irene Machura**[4], **Meshack Liru**[5], **Joyce Akuka**[6], **Deborah Omedo**[4], **Samwel Symekher**[7], **Samoel A. Khamadi**[7], **Lynda Isaaka**[8], **Morris Ogero**[8], **Livingstone Mumelo**[8], **James A. Berkley**[8,9,10,11], **Ambrose Agweyu**[8,11], **Judd L. Walson**[9,12], **Benson O. Singa**[3], **Kirkby D. Tickell**[2,9]

1 Department of Epidemiology, University of Washington, Seattle, Washington, United States of America,
2 Departments of Global Health, University of Washington, Seattle, Washington, United States of America,
3 Center for Clinical Research, Kenya Medical Research Institute, Nairobi, Kenya, 4 Kisii Teaching and Referral Hospital, Kisii, Kenya, 5 Homa Bay County Referral Hospital, Homa Bay, Kenya, 6 Migori County Referral Hospital, Migori, Kenya, 7 Center for Virus Research, Kenya Medical Research Institute, Nairobi, Kenya, 8 KEMRI/Wellcome Trust Research Programme, Nairobi, Kenya, 9 The Childhood Acute Illness and Nutrition Network (CHAIN), Nairobi, Kenya, 10 Centre for Tropical Medicine & Global Health Nuffield Department of Medicine, University of Oxford, Oxford, United Kingdom, 11 Department of Infectious Disease Epidemiology, London School of Hygiene & Tropical Medicine, London, United Kingdom, 12 Departments of Global Health, Medicine (Infectious Disease), Pediatrics and Epidemiology, University of Washington, Seattle, Washington, United States of America

* adino@uw.edu

**Data Availability Statement:** All data in Supporting information files.

**Funding:** JLW received the grant from Bill & Melinda Gates Foundation [INV016894] (https://

## Abstract

The epidemiology of pediatric COVID-19 in sub-Saharan Africa and the role of fecal-oral transmission in SARS-CoV-2 are poorly understood. Among children and adolescents in Kenya, we identify correlates of COVID-19 infection, document the clinical outcomes of infection, and evaluate the prevalence and viability of SARS-CoV-2 in stool. We recruited a prospective cohort of hospitalized children aged two months to 15 years in western Kenya between March 1 and June 30 2021. Children with SARS-CoV-2 were followed monthly for 180-days after hospital discharge. Bivariable logistic regression analysis was used to identify the clinical and sociodemographics correlates of SARS-CoV-2 infection. We also calculated the prevalence of SARS-CoV-2 detection in stool of confirmed cases. Of 355 systematically tested children, 55 (15.5%) were positive and were included in the cohort. The commonest clinical features among COVID-19 cases were fever (42/55, 76%), cough (19/55, 35%), nausea and vomiting (19/55, 35%), and lethargy (19/55, 35%). There were no statistically significant difference in baseline sociodemographic and clinical characteristics between SARS-CoV-2 positive and negative participants. Among positive participants, 8/55 (14.5%, 95%CI: 5.3%-23.9%) died; seven during the inpatient period. Forty-nine children with COVID-19 had stool samples or rectal swabs available at baseline, 9 (17%) had PCR-positive stool or rectal swabs, but none had SARS-CoV-2 detected by culture. Syndromic identification of COVID-19 is particularly challenging among children as the presenting symptoms and signs mirror other common pediatric diseases. Mortality among children hospitalized with COVID-19 was high in this cohort but was comparable to mortality seen with other common illnesses in this setting. Among this small set of children with COVID-19 we

www.gatesfoundation.org/). The funders had no role in study design, data collection and analysis, decision to publish, or preparation of the manuscript.

**Competing interests:** The authors have declared that no competing interests exist.

detected SARS-CoV-2 DNA, but were not able to culture viable SARs-CoV-2 virus, in stool. This suggests that fecal transmission may not be a substantial risk in children recently diagnosed and hospitalized with COVID-19 infection.

## Background

Severe acute respiratory syndrome coronavirus 2 (SARS-CoV-2) has caused millions of hospitalizations and deaths [1, 2] across the world. In Kenya, the first official case of COVID-19 was reported in March 2020 and by the end of November 2022, more than 341,235 people had been infected, with 5,684 deaths. However, from these data, <1% of the deaths occurred among children under the age of 15 [3]. Data from high income countries suggests that most COVID-19 cases among children and adolescents are asymptomatic or mild, although severe disease, including multisystem inflammatory syndrome (MIS), may develop [4–6]. In resource-limited settings, where children have a higher prevalence of comorbidities, including malnutrition, HIV and other infections, COVID-19 management maybe more complicate and may have worse clinical outcomes [7]. However, there are limited data on the epidemiologic profile, diagnosis and risk factors for COVID-19 among pediatric patients in resource-limited settings.

Fever and cough are the most frequently reported clinical features in pediatric COVID-19, but diarrhea is also a common manifestation [8–10]. The symptomatic overlap of COVID-19 with other prevalent diseases in resource-limited settings, including diarrheal disease, malaria and pneumonia, may make the syndromic identification of COVID-19 challenging [11–14]. Polymerase Chain Reaction (PCR) testing from nasopharyngeal samples or saliva is the gold standard for COVID-19 diagnosis but is often unavailable or not systematically collected in acutely ill children in low resource settings. In addition, several studies have reported the detection of SARs-CoV-2 in stool samples [15–17], highlighting that feco-oral transmission may be an important mode of transmission, particularly in low resource settings where access to adequate sanitation and clean water may be limited [18–20].

We sought to identify clinical and sociodemographic correlates of COVID-19 among hospitalized children in Kenya and to evaluate PCR and culture-based detection of SARS-CoV-2 in fecal samples as a measure of the potential for fecal transmission.

## Methods

### Study design, setting, and population

This study was conducted during the third wave of COVID-19 in Kenya when the B.1.1.7 (*Alpha*) was the dominant variant [21, 22]. We systematically tested all children admitted to three hospitals in Kenya (Homa Bay County Referral, Kisii Teaching and Referral, and Migori County Referral hospitals), who were aged two months to 15 years for SARs CoV-2 between March 1 and June 30, 2021. Systematic testing excluded suspected COVID-19 cases who had already been at the study hospital for more than 24 hours or were referred after 24 hours of admission to another hospital, as these children may have acquired the infection during their hospitalization. Baseline clinical and sociodemographic data were collected from all patients prior to SARs CoV-2 testing.

Confirmed cases were enrolled in a prospective cohort study of hospitalized children aged two months to 15 years if they planned to remain within the hospital catchment area for at least six months, and consented to participate in the study. Cohort participants were followed daily during hospitalization and monthly for 6-months after hospital discharge.

Participants were scheduled for monthly follow-up visits in the outpatient clinics, unless the participant was hospitalized at the time of the visit. When in-person visits were not possible, interviews were conducted through a phone call and participants' vital status was recorded. Anthropometric measurements were only done at in-person follow-up visits. Follow-up ended when the child completed their six-month follow-up visit or when a child died. Families that missed the 6-month visit and were not traceable via phone calls or home visits were declared lost to follow up. Nasopharyngeal swab, rectal swab, stool, and blood samples were taken at admission.

## Study outcomes

The primary outcomes of this study were the clinical profile and correlates of SARS-CoV-2 infection and its detection by PCR and culture in stool or rectal swab samples. Secondary outcomes include the mortality rate among cohort participants.

## Statistical analysis

Descriptive statistics summarizes distribution of variables between confirmed COVID-19 cases and children who were not infected. A chi-square test was used to evaluate the relationship between COVID-19 status and categorical variables, and a t-test was used for continuous variables. We selected possible clinical and sociodemographic correlates of SARS-CoV-2 infection a priori based on previously published literature [21–23]. The clinical variables were sickle cell disease, fever, lethargy, vomiting/nausea, difficulty of breathing, convulsion, diarrhea, cough, headache, not feeding, abdominal pain, fatigue, reduced air entry, wheeze, chest indrawing, crackles, capillary refill, activity (mental status of being irritable/Agitated or lethargic), and jaundice. The sociodemographic characteristics were age, sex, nutritional status (wasting measured by mid-upper arm circumference (MUAC), weight for height Z score (WHZ), and body mass index (BMI), and stunting), breastfeeding, birth order, caregiver marital status, and caregiver educational status. We used logistic regression to evaluate the association of each variable with SARS-CoV-2 infection. In addition, we explored the frequency of mortality and its distribution across key clinical correlates.

## Materials and techniques for stool sample processing

SARS-CoV-2 was detected in stool and rectal swab samples using PCR and viral culture. Initially, 180-220mg of swab /stool was placed in an Eppendorf tube, where 1 ml of the swab/stool lysis buffer was added to the fecal sample and was further processed. Quantitative Real-Time PCR (qRT-PCR) was carried out using primers and probes from the Da An SARSCoV-2 detection kit. This kit detects the presence of SARS-CoV-2 ORF-1ab and N genes explicitly. The real-time PCR was repeated using another kit, specifically testing the 'reactive' sample. In addition, the PCR-positive samples were further processed and inoculated to the cell line (Vero-E6) for isolation and detection of a live virus and followed up for a week at 35˚C. To reliably determine the isolation of the virus from fecal samples, the samples were re-inoculated in the same cell lines and incubated for an additional 10–12 days. RNA was extracted using the RADI Nucleic Acid Extraction protocol and tested for the presence of the virus using the Sansure Biotech Novel Coronavirus (2019-nCoV) diagnostic kit. The kit contains specific primers and probes for detecting the SARS-CoV-2 largest gene (ORF1ab) and the nucleocapsid (N) gene regions. Where both whole stool and rectal swabs were available, both sample types were tested to maximize the chance of isolating live fecal virus [23].

### Ethics

Ethical approval was obtained from the University of Washington Insitutional Review Board and all participating institutions. Written consent was obtained from caregivers of all study participants.

## Results

A total of 355 children screened, and 55 (16%) were SARS CoV-2 positive and enrolled in the cohort. In addition, 296 (83%) tested negative for SARS CoV-2 and 4 (1%) had an inconclusive test result (Fig 1).

Most participants were over two years old (216, 61%), a quarter (80, 23%) had severe wasting, 49 (14%) had severe stunting, and 6 (2%) were HIV infected. Among the 139 children aged less than 2 years, 71 (51%) were breastfeeding, and out of 28 children aged less than 6 months, 7 (25%) were exclusively breastfeeding (Table 1).

### Clinical features of COVID-19

The most frequent clinical feature among COVID-19 infected children was fever (42, 76%), followed by cough (19, 35%), nausea and vomiting (19, 35%), and lethargy (19, 35%). The

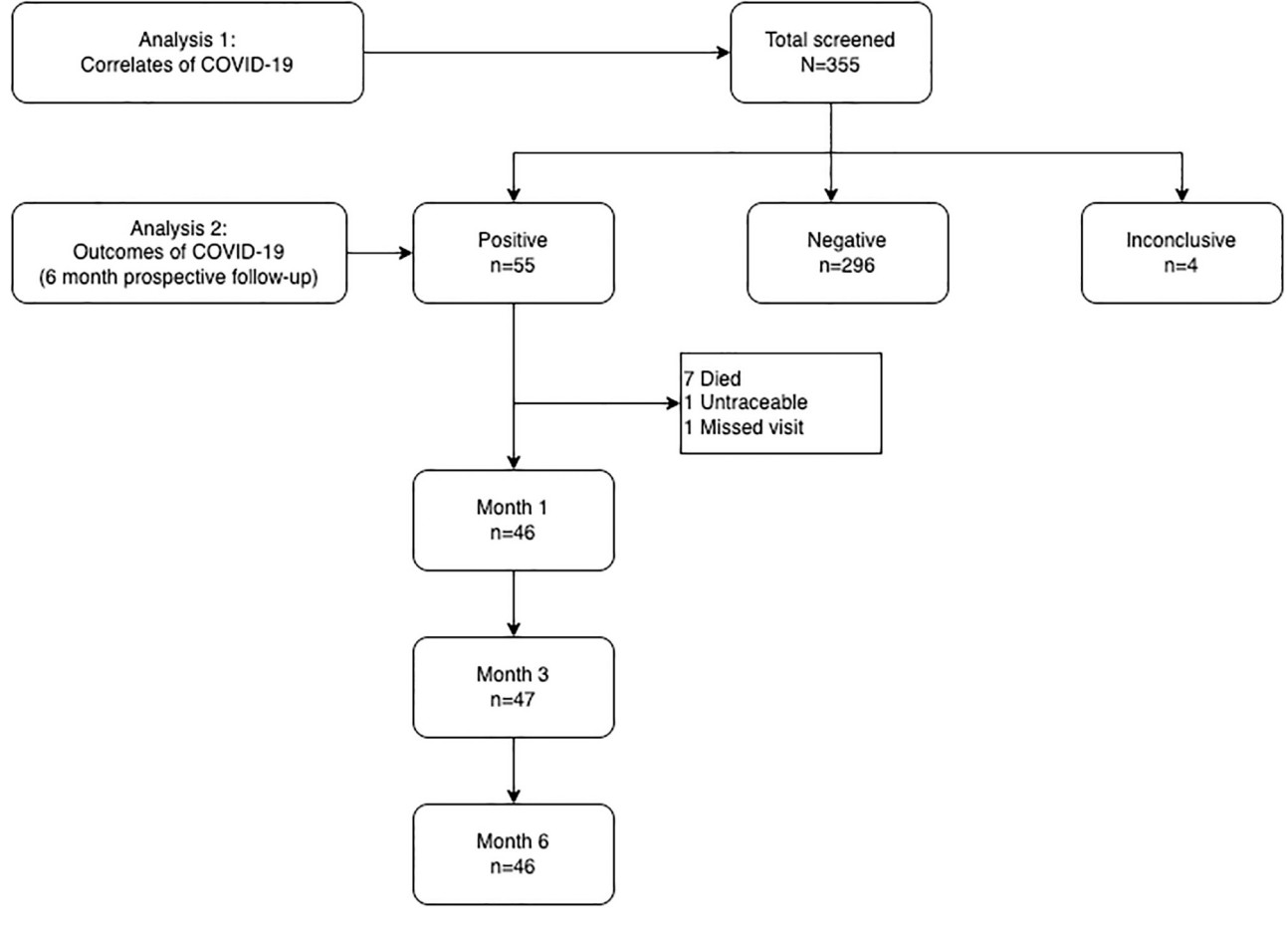

**Fig 1. Flow chart.**

**Table 1. Baseline characteristics of participants.**

| Characteristic | Overall, N = 355[1] | COVID-19 Negative, N = 300[1] | COVID-19 Positive, N = 55[1] |
|---|---|---|---|
| **Study Hospital** | | | |
| Homa Bay | 42 (12%) | 27 (9.0%) | 15 (27%) |
| Kisii | 85 (24%) | 73 (24%) | 12 (22%) |
| Migori | 228 (64%) | 200 (67%) | 28 (51%) |
| **Age** | | | |
| <6 months | 28 (7.9%) | 25 (8.3%) | 3 (5.5%) |
| 6–11 months | 40 (11%) | 32 (11%) | 8 (15%) |
| 12–23 months | 71 (20%) | 58 (19%) | 13 (24%) |
| 2–5 years | 123 (35%) | 108 (36%) | 15 (27%) |
| > = 5 years | 93 (26%) | 77 (26%) | 16 (29%) |
| **Sex** | | | |
| Female | 161 (45%) | 130 (43%) | 31 (56%) |
| Male | 194 (55%) | 170 (57%) | 24 (44%) |
| **Nutritional status** | | | |
| No Malnutrition | 230 (65%) | 190 (64%) | 40 (73%) |
| MAM | 43 (12%) | 40 (13%) | 3 (5.5%) |
| SAM | 80 (23%) | 68 (23%) | 12 (22%) |
| Unknown | 2 | 2 | 0 |
| **Stunting** | | | |
| Severe stunting | 49 (14%) | 42 (14%) | 7 (13%) |
| Moderate stunting | 41 (12%) | 35 (12%) | 6 (11%) |
| No stunting | 255 (74%) | 214 (74%) | 41 (76%) |
| Unknown | 10 | 9 | 1 |
| **Breastfeeding status** | | | |
| Exclusive breastfeeding | 66 (19%) | 56 (19%) | 10 (18%) |
| Non-exclusive breastfeeding | 11 (3.1%) | 7 (2.3%) | 4 (7.3%) |
| Not breastfeeding | 278 (78%) | 237 (79%) | 41 (75%) |
| **Cerebral palsy** | 19 (5.4%) | 17 (5.7%) | 2 (3.6%) |
| **Sickle cell disease** | 18 (5.1%) | 14 (4.7%) | 4 (7.3%) |
| **HIV infection** | 6 (1.7%) | 6 (2.0%) | 0 (0%) |
| **Birth order** | | | |
| First | 83 (23%) | 70 (23%) | 13 (24%) |
| Second | 87 (25%) | 71 (24%) | 16 (29%) |
| Third and above | 185 (52%) | 159 (53%) | 26 (47%) |
| **Caregiver age** | | | |
| <18 Years | 1 (0.3%) | 1 (0.3%) | 0 (0%) |
| > = 18 Years | 339 (95%) | 286 (95%) | 53 (96%) |
| >50 Years | 13 (3.7%) | 11 (3.7%) | 2 (3.6%) |
| Care home or unclear | 2 (0.6%) | 2 (0.7%) | 0 (0%) |
| **Caregiver marital status** | | | |
| Married | 295 (83%) | 250 (83%) | 45 (82%) |
| Unmarried | 59 (17%) | 49 (16%) | 10 (18%) |
| NA | 1 (0.3%) | 1 (0.3%) | 0 (0%) |
| **Caregiver educational status** | | | |
| None | 21 (5.9%) | 16 (5.3%) | 5 (9.1%) |
| Primary | 215 (61%) | 180 (60%) | 35 (64%) |
| Secondary | 87 (25%) | 77 (26%) | 10 (18%) |

(*Continued*)

**Table 1.** (Continued)

| Characteristic | Overall, N = 355[1] | COVID-19 Negative, N = 300[1] | COVID-19 Positive, N = 55[1] |
|---|---|---|---|
| Above secondary | 30 (8.5%) | 25 (8.3%) | 5 (9.1%) |
| Unknown/NA | 2 (0.6%) | 2 (0.7%) | 0 (0%) |
| **Primary caregiver** | | | |
| Biological Mother | 324 (91%) | 272 (91%) | 52 (95%) |
| Other | 31 (8.7%) | 28 (9.3%) | 3 (5.5%) |
| **Known household COVID-19 contact** | 6 (1.7%) | 6 (2.0%) | 0 (0%) |
| **Respiratory rate** | 34 ± 9 | 34 ± 9 | 33 ± 9 |
| **Heart rate** | 123 ± 24 | 123 ± 23 | 121 ± 25 |
| **Temperature** | 36.8 ± 0.7 | 36.8 ± 0.8 | 36.7 ± 0.5 |
| **Oxygen saturation** | | | |
| <90% | 31 (8.7%) | 24 (8.0%) | 7 (13%) |
| > = 90% | 324 (91%) | 276 (92%) | 48 (87%) |

[1]n (%); Mean (SD)

distribution of baseline sociodemographic and clinical characteristics was not statistically significantly different between COVID-19 infected and non-infected participants.

In the bivariable analysis, we observed a lower risk (marginally non-significant) of COVID-19 among males as compared to females (Crude Odds Ratio (COR): 0.59 (95%CI: 0.33, 1.05), p-value = 0.075) and among those who reported a headache compared to those who did not (COR: 0.34 (95%CI: 0.08, 0.98), p-value = 0.08) (Table 2).

### Outcomes of the cohort

In the cohort, 8/55 (15%) of the participants died before the end of the 180-day follow-up period, with seven of the eight deaths occurring prior to hospital discharge (7/55–12.7% [95% CI: 5.3, 24.5%]). The remaining death occurred after the third month of follow-up (Fig 2). Four of the eight children who died were below 2 years of age at enrolment and three were above five years of age. Of the children who died, five had severe acute malnutrition, three had severe stunting and one had cerebral palsy. At admission, four were lethargic, three were not feeding, four had a fever, three had a cough, and three had nausea and vomiting.

### Fecal shedding of SARS-CoV-2

Of the 55 children with COVID-19, 53 had stool and/or rectal swabs collected at enrolment; 49 children had a rectal swab and 38 had whole stool samples. Overall, 9 (17%) of these children had a positive PCR for COVID-19 from stool or rectal swab (Table 3), with 5 (10%) positive results from rectal swabs and 8 (21%) from whole stool. Four children had SARS-CoV-2 detected in both sample types at enrolment. None of the PCR-positive rectal swabs or stool samples was culture positive, despite repeat attempts to culture both stool and rectal swabs. The clinical profiles of children with stool and/or rectal samples collected are given by PCR result in Table 3.

### Discussion

In western Kenya, fever, cough, nausea and vomiting, and lethargy were the most frequent signs and symptoms observed among children testing positive for COVID-19. However, there was no difference in the sociodemographic and clinical characteristics of children with

**Table 2. Logistic regression table for the correlates of COVID-19 infection.**

| Characteristic | N | 0, N = 296[1] | 1, N = 55[1] | COR[2] | 95% CI[2] | p-value |
|---|---|---|---|---|---|---|
| **Age** | 351 | | | | | |
| <6 months | | 25 (8.4%) | 3 (5.5%) | — | — | |
| 6–11 months | | 31 (10%) | 8 (15%) | 2.15 | 0.56, 10.6 | 0.3 |
| 12–23 months | | 57 (19%) | 13 (24%) | 1.9 | 0.55, 8.81 | 0.3 |
| 2–5 years | | 107 (36%) | 15 (27%) | 1.17 | 0.35, 5.32 | 0.8 |
| > = 5 years | | 76 (26%) | 16 (29%) | 1.75 | 0.53, 7.98 | 0.4 |
| **Sex** | 351 | | | | | |
| Female | | 128 (43%) | 31 (56%) | — | — | |
| Male | | 168 (57%) | 24 (44%) | 0.59 | 0.33, 1.05 | 0.075 |
| **Birth order** | 351 | | | | | |
| First | | 70 (24%) | 13 (24%) | — | — | |
| Second | | 70 (24%) | 16 (29%) | 1.23 | 0.55, 2.79 | 0.6 |
| Third and above | | 156 (53%) | 26 (47%) | 0.9 | 0.44, 1.90 | 0.8 |
| **Caregiver marital status** | 351 | | | | | |
| Married | | 246 (83%) | 45 (82%) | — | — | |
| Unmarried | | 50 (17%) | 10 (18%) | 1.09 | 0.49, 2.24 | 0.8 |
| **Caregiver educational status** | 350 | | | | | |
| Above secondary | | 24 (8.1%) | 5 (9.1%) | — | — | |
| None | | 16 (5.4%) | 5 (9.1%) | 1.5 | 0.36, 6.23 | 0.6 |
| Primary | | 180 (61%) | 35 (64%) | 0.93 | 0.36, 2.92 | 0.9 |
| Secondary | | 75 (25%) | 10 (18%) | 0.64 | 0.21, 2.22 | 0.5 |
| **Nutritional status** | 349 | | | | | |
| No Malnutrition | | 189 (64%) | 40 (73%) | — | — | |
| MAM | | 40 (14%) | 3 (5.5%) | 0.35 | 0.08, 1.04 | 0.1 |
| SAM | | 65 (22%) | 12 (22%) | 0.91 | 0.44, 1.81 | 0.8 |
| **Breastfeeding** | 351 | | | | | |
| Exclusive breastfeeding | | 56 (19%) | 10 (18%) | — | — | |
| Non-exclusive breastfeeding | | 7 (2.4%) | 4 (7.3%) | 3.2 | 0.73, 12.8 | 0.1 |
| Non-breast feeding | | 233 (79%) | 41 (75%) | 0.99 | 0.48, 2.19 | >0.9 |
| **Known sickle cell disease** | 351 | 14 (4.7%) | 4 (7.3%) | 1.58 | 0.43, 4.61 | 0.4 |
| **Fever** | 351 | 228 (77%) | 42 (76%) | 0.96 | 0.50, 1.96 | >0.9 |
| **Headache** | 351 | 43 (15%) | 3 (5.5%) | 0.34 | 0.08, 0.98 | 0.08 |
| **Vomiting/Nausea** | 351 | 81 (27%) | 19 (35%) | 1.4 | 0.75, 2.56 | 0.3 |
| **Diarrhea <14 days** | 351 | 46 (16%) | 9 (16%) | 1.06 | 0.46, 2.23 | 0.9 |
| **Abdominal pain** | 142 | 54 (43%) | 4 (24%) | 0.4 | 0.11, 1.22 | 0.13 |
| **Not feeding** | 351 | 19 (6.4%) | 7 (13%) | 2.13 | 0.79, 5.13 | 0.11 |
| **Cough <14 days** | 351 | 113 (38%) | 19 (35%) | 0.85 | 0.46, 1.55 | 0.6 |
| **Difficulty of breathing** | 153 | 59/132 (45%) | 8/21 (38%) | 0.76 | 0.28, 1.93 | 0.6 |
| **Fatigue** | 351 | 57 (19%) | 12 (22%) | 1.17 | 0.56, 2.30 | 0.7 |
| **Lethargy** | 351 | 98 (33%) | 19 (35%) | 1.07 | 0.57, 1.94 | 0.8 |
| **Convulsion** | 351 | 31 (10%) | 5 (9.1%) | 0.85 | 0.28, 2.13 | 0.8 |
| **Activity** | 351 | | | | | |
| Irritable/Agitated | | 22 (7.4%) | 4 (7.3%) | — | — | |
| Lethargic | | 47 (16%) | 12 (22%) | 1.4 | 0.43, 5.47 | 0.6 |
| Normal | | 227 (77%) | 39 (71%) | 0.94 | 0.34, 3.36 | >0.9 |
| **Jaundice** | 351 | 38 (13%) | 3 (5.5%) | 0.39 | 0.09, 1.14 | 0.13 |
| **Reduced air entry** | 351 | 27 (9.1%) | 5 (9.1%) | 1 | 0.33, 2.51 | >0.9 |

(*Continued*)

**Table 2.** (Continued)

| Characteristic | N | 0, N = 296[1] | 1, N = 55[1] | COR[2] | 95% CI[2] | p-value |
|---|---|---|---|---|---|---|
| **Wheeze** | 351 | 37 (12%) | 4 (7.3%) | 0.55 | 0.16, 1.44 | 0.3 |
| **Chest indrawing** | 351 | 28 (9.5%) | 7 (13%) | 1.4 | 0.54, 3.22 | 0.5 |
| **Crackles** | 351 | 38 (13%) | 4 (7.3%) | 0.53 | 0.15, 1.40 | 0.2 |
| **Capillary refill** | 351 | | | | | |
| <2 Seconds | | 138 (47%) | 30 (55%) | — | — | |
| 2–3 Seconds | | 133 (45%) | 21 (38%) | 0.73 | 0.39, 1.33 | 0.3 |
| >3 Seconds | | 25 (8.4%) | 4 (7.3%) | 0.74 | 0.21, 2.07 | 0.6 |
| **Hemoglobin** | 334 | 8.54+/-2.77 | 9.27+/-2.90 | 1.1 | 0.99, 1.22 | 0.08 |
| **WBC** | 334 | 15+/-11 | 13+/-10 | 0.99 | 0.96, 1.02 | 0.4 |
| **RBC** | 333 | 3.49+/-1.33 | 3.80+/-1.28 | 1.19 | 0.96, 1.48 | 0.11 |
| **Platelets** | 332 | 244+/-187 | 216+/-148 | 1 | 1.00, 1.00 | 0.3 |

[1]n (%)

[2]OR = Odds Ratio, CI = Confidence Interval

confirmed COVID-19 in comparison to those children who tested negative for COVID-19. This finding highlighting the challenges clinicians face in identifying children with COVID-19 in the absence of laboratory confirmatory testing, particularly given the overlap of these non-specific symptoms with other common illnesses frequently presenting to hospital [24–30].

In this study, 14% of the children with COVID-19 died, and all but one death occurred during initial hospitalization. A recent study by Nachega et al. in six African countries, reported that 8.3% of children with COVID-19 died, while a similar study from Brazil among children

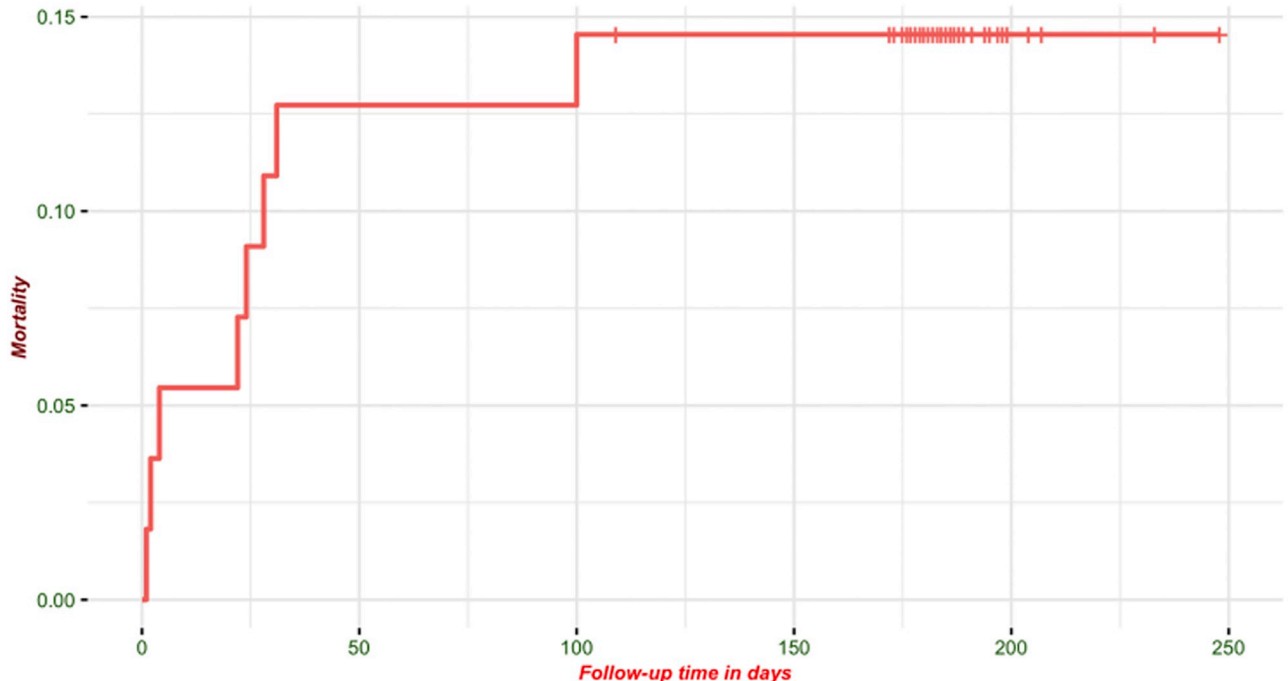

**Fig 2. Kaplan Meier survival curve of children with confirmed COVID-19.**

**Table 3. Differences in clinical profile between stool/rectal swab positive and negative participants.**

| Characteristic | N | Stool/rectal swab PCR negative, N = 44[1] | Stool/rectal swab PCR positive, N = 9[1] |
|---|---|---|---|
| **General clinical characteristics** | | | |
| Fever | 53 | 35 (80%) | 7 (78%) |
| Fatigue | 53 | 8 (18%) | 4 (44%) |
| Headache | 53 | 3 (6.8%) | 0 (0%) |
| Not feeding | 53 | 5 (11%) | 1 (11%) |
| Oxygen saturation | 53 | 94.9 +/-7.2 | 98.0 +/-1.4 |
| Temperature | 53 | 36.72 +/-0.50 | 36.68 +/-0.34 |
| **Respiratory** | | | |
| Cough <14 days | 53 | 17 (39%) | 0 (0%) |
| Difficulty of breathing | 20 | 7 (41%) | 0 (0%) |
| Chest indrawing | 53 | 6 (14%) | 0 (0%) |
| Reduced air entry | 53 | 4 (9.1%) | 0 (0%) |
| Wheeze | 53 | 4 (9.1%) | 0 (0%) |
| Crackles | 53 | 4 (9.1%) | 0 (0%) |
| Respiratory rate | 53 | 33 +/-9 | 29 +/-4 |
| **Gastrointestinal** | | | |
| Diarrhea <14 days | 53 | 7 (16%) | 1 (11%) |
| Vomiting/Nausea | 53 | 17 (39%) | 1 (11%) |
| Abdominal pain | 17 | 4 (29%) | 0 (0%) |
| Jaundice | 53 | 3 (6.8%) | 0 (0%) |
| **Neurological symptoms** | | | |
| Lethargy | 53 | 15 (34%) | 3 (33%) |
| Convulsion | 53 | 5 (11%) | 0 (0%) |
| **Activity** | 53 | | |
| Irritable/Agitated | | 2 (4.5%) | 2 (22%) |
| Lethargic | | 8 (18%) | 3 (33%) |
| Normal | | 34 (77%) | 4 (44%) |
| **Circulatory** | | | |
| **Capillary refill** | 53 | | |
| <2 Seconds | | 28 (64%) | 2 (22%) |
| 2–3 Seconds | | 13 (30%) | 7 (78%) |
| >3 Seconds | | 3 (6.8%) | 0 (0%) |
| Heart rate | 53 | 119 +/-26 | 132 +/-21 |
| Hemoglobin | 53 | 9.43 +/-2.98 | 8.68 +/-2.31 |
| WBC | 53 | 12 +/-6 | 23 +/-18 |
| RBC | 53 | 3.87 +/-1.30 | 3.54 +/-1.02 |
| Platelet | 53 | 214 +/-147 | 255 +/-152 |

[1]n (%)

and adolescents below age <20 reported a 7.6% mortality rate during hospital admission [24, 31]. These findings are consistent with the mortality rate observed in our study. Importantly, five of the eight deaths observed were among children with severe wasting. Children with COVID-19 appear to have similar mortality risk to children with other common medical conditions resulting in hospital admission in sub-Saharan Africa [32, 33].

Evidence from high-income countries suggests that COVID-19-related mortality among hospitalized children and adolescents is less than 5% [28, 34–36], considerably lower than that

observed in this study. This disparity is likely to be related to higher capacity of health facilities to care for acute unwell children and the lower prevalence of comorbidities such as wasting and stunting in high income countries. Efforts to address existing comorbidities and strengthen health systems in low resource settings are urgently needed to reduce mortality, not only from COVID-19 but from other common pediatric conditions observed in these settings.

Only a quarter (27%) of children testing positive for COVID-19 by PCR from nasopharyngeal swabs had PCR-detectable SARS-CoV-2 in either stool or rectal swabs. Despite evidence that SARS-CoV-2 can infect the intestine and cause inflammation and gastrointestinal symptoms, children in this study with positive stool PCR tests did not have more reported diarrhea than COVID-19 negative children [37–39]. A recent systematic review and meta-analysis evaluated the performance of COVID-19 PCR detection in adults and reported that stool sample or rectal swab PCR tests have a sensitivity of 24% (16.7%, 33.0%), as opposed to a sensitivity of 73% when using nasopharyngeal swab samples [16]. A similar meta analysis that included children and adults, found that 25% of individuals with positive nasopharyngeal samples had a positive stool PCR test, and that this proportion did not differ between age groups [18]. In contrast, a third systematic review reported that SARS-CoV-2 could be detected in 75% of stool samples collected from children with COVID-19 [40]. There is clearly heterogeneity in the proportion of COVID-19 cases who have PCR-detectable SARS-CoV-2 in their stool, which may be related to timing of sample collection relative to infection or severity of infection, variant of virus, or other host factors. Interestingly, despite repeated efforts using both whole stool and rectal swabs, we were not able to culture viable virus from any fecal sample in this study. Similarly low isolation rates have been observed in studies form China, Germany, Korea and Singapore [41, 42], suggesting that across these diverse settings there may be limited viable virus shed in stool and that fecal shedding of SARS-CoV-2 may not be a major route of transmission.

This study had notable strengths, particularly the systematic screening for SARs CoV-2, the long duration of follow-up, and the extensive efforts to culture live virus from stool. However, the study had a relatively small sample size and the observational nature of the study limits our ability to infer causality. Secondly, the study was conducted in a hospital setting and these results may not be generalizable to children living in the community. Finally, due to isolation protocols and a high rate of early mortality, we were unable collect whole stool or rectal swab samples on a small number of children, which may have introduced a degrees of selection bias.

## Conclusion

The non-specific nature of the signs and symptoms associated with COVID-19 in children make it difficult to differentiate from other common pediatric infections in sub-Saharan Africa. This poses a particular challenge in resource-limited settings where diagnostics are often unavailable or unaffordable. Mortality among children hospitalized with COVID-19 was high, but appears comparable to mortality observed among children hospitalized for other causes in the region. SARs CoV-2 was only detected in stool or rectal swabs by PCR in a quarter of children with confirmed COVID- 19 infection and live virus was not detected in any fecal samples. These data suggest that fecal transmission may not play a substantial role in the transmission of the virus.

## Supporting information

**S1 Text. Completed inclusivity questionnaire.**
(DOCX)

**S1 Data. Data underlying the study findings.**
(CSV)

**S1 Checklist. Completed STROBE checklist.**
(DOCX)

## Acknowledgments

The authors would like to acknowledge the support of the Bill & Melinda Gates Foundation (INV016894), and the participants of this research. For the purpose of open access, a Creative Commons Attribution is applied under the grant conditions of the Foundation for any author-accepted manuscript version arising from this submission.

## Author Contributions

**Conceptualization:** Christina Sherry, Chrisantus Oduol, James A. Berkley, Ambrose Agweyu, Judd L. Walson, Benson O. Singa, Kirkby D. Tickell.

**Data curation:** Adino Tesfahun Tsegaye, Mary Masheti, Samwel Symekher, Samoel A. Khamadi, Lynda Isaaka, Morris Ogero, Livingstone Mumelo.

**Formal analysis:** Adino Tesfahun Tsegaye.

**Funding acquisition:** James A. Berkley, Ambrose Agweyu, Judd L. Walson, Benson O. Singa, Kirkby D. Tickell.

**Investigation:** Chrisantus Oduol, Joyce Otieno, Doreen Rwigi, Mary Masheti, Irene Machura, Meshack Liru, Joyce Akuka, Samwel Symekher, Samoel A. Khamadi, Lynda Isaaka, Morris Ogero, Livingstone Mumelo, James A. Berkley, Ambrose Agweyu, Judd L. Walson, Benson O. Singa, Kirkby D. Tickell.

**Methodology:** Adino Tesfahun Tsegaye, Christina Sherry, Doreen Rwigi, Mary Masheti, Samwel Symekher, Samoel A. Khamadi, James A. Berkley, Ambrose Agweyu, Judd L. Walson, Benson O. Singa, Kirkby D. Tickell.

**Project administration:** Adino Tesfahun Tsegaye, Chrisantus Oduol, Joyce Otieno, Doreen Rwigi, Mary Masheti, Irene Machura, Meshack Liru, Joyce Akuka, Deborah Omedo, Lynda Isaaka, Morris Ogero, Livingstone Mumelo, James A. Berkley, Ambrose Agweyu, Judd L. Walson, Benson O. Singa, Kirkby D. Tickell.

**Resources:** Irene Machura, James A. Berkley, Ambrose Agweyu, Judd L. Walson, Kirkby D. Tickell.

**Software:** Adino Tesfahun Tsegaye, Kirkby D. Tickell.

**Supervision:** Christina Sherry, Chrisantus Oduol, Joyce Otieno, Doreen Rwigi, Mary Masheti, Meshack Liru, Joyce Akuka, Lynda Isaaka, James A. Berkley, Ambrose Agweyu, Judd L. Walson, Benson O. Singa, Kirkby D. Tickell.

**Validation:** Adino Tesfahun Tsegaye.

**Visualization:** Adino Tesfahun Tsegaye.

**Writing – original draft:** Adino Tesfahun Tsegaye.

**Writing – review & editing:** Adino Tesfahun Tsegaye, James A. Berkley, Ambrose Agweyu, Judd L. Walson, Kirkby D. Tickell.

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
