## [Decision Letter · Decision Letter 0]

11 Apr 2023

PGPH-D-23-00293

Clinical epidemiology of COVID-19 among hospitalized children in rural western Kenya

Dear  Corresponding Author,

Thank you for submitting your manuscript to PLOS Global Public Health. After careful consideration, we feel that it has merit but does not fully meet PLOS Global Public Health’s publication criteria as it currently stands. Therefore, we invite you to submit a revised version of the manuscript that addresses the points raised during the review process.

We look forward to receiving your revised manuscript.

Kind regards,

Shivanthi Samarasinghe, PhD

Academic Editor

Journal Requirements:

Additional Editor Comments (if provided):

Reviewers' comments:

Reviewer's Responses to Questions

**Comments to the Author**

1. Does this manuscript meet PLOS Global Public Health’s publication criteria? Is the manuscript technically sound, and do the data support the conclusions? The manuscript must describe methodologically and ethically rigorous research with conclusions that are appropriately drawn based on the data presented.

Reviewer #1: Yes

2. Has the statistical analysis been performed appropriately and rigorously?

Reviewer #1: Yes

3. Have the authors made all data underlying the findings in their manuscript fully available (please refer to the Data Availability Statement at the start of the manuscript PDF file)?

Reviewer #1: Yes

4. Is the manuscript presented in an intelligible fashion and written in standard English?

Reviewer #1: Yes

5. Review Comments to the Author

Reviewer #1: Tsegaye et al. describe the clinical characteristics and associations with COVID-19 among hospitalized children near Kisumu, Kenya. This well-written manuscript represents multiple important contributions, including lack of culturable SARS-CoV-2 in stool from hospitalized children, and similar demographic and clinical characteristics in hospitalized children regardless of SARS-CoV-2 status.

A CONSORT diagram could help to show who was included and excluded from analysis for different reasons, and who was followed-up to ascertain survival.

It appears only a small fraction (53 out of >300) children were able to be followed-up to ascertain outcomes. Could the authors comment on potential biases due to the strong down-selection of who was followed-up?

At the end of the paper, the authors note that survival in the SARS-CoV-2 PCR+ cohort is similar to survival rates reported in SARS-CoV-2 negative children admitted to hospitals in this region. Do the authors have access to SARS-CoV-2 negative or overall child survival data in order to make a comparison? It would be interesting to know if observed mortality rates differ by SARS-CoV-2 status, and this might enable the authors to make a stronger point about children dying of other causes.

These are just minor suggestions – overall, very clear and interesting work!

6. PLOS authors have the option to publish the peer review history of their article (what does this mean?). If published, this will include your full peer review and any attached files.

**Do you want your identity to be public for this peer review?** For information about this choice, including consent withdrawal, please see our Privacy Policy.

Reviewer #1: **Yes: **Anna Bershteyn

---

## [Editor Report · Decision Letter 1]

22 May 2023

Clinical epidemiology of COVID-19 among hospitalized children in rural western Kenya

PGPH-D-23-00293R1

Dear Corresponding Author,

The manuscript describes the clinical characteristics and associations with COVID-19 among hospitalized children near Kisumu, Kenya, which is relevant and timely to the scope of the journal.

We are pleased to inform you that your manuscript 'Clinical epidemiology of COVID-19 among hospitalized children in rural western Kenya' has been provisionally accepted with minor revisions for publication in PLOS Global Public Health.

Before your manuscript can be formally accepted you will need to complete the revisions raised by the reviewer,  some formatting changes, which you will receive in a follow up email. A member of our team will be in touch with a set of requests.

Best regards,

Shivanthi Samarasinghe, PhD

Academic Editor